# The Association between Perceived Stress, Quality of Life, and Level of Physical Activity in Public School Teachers

**DOI:** 10.3390/ijerph21010088

**Published:** 2024-01-12

**Authors:** Claudiele Carla Marques da Silva, Amanda Barbosa dos Santos, Isabella Cristina Leoci, Enrique Gervasoni Leite, Ewerton Pegorelli Antunes, Wesley Torres, Eduardo Duarte de Lima Mesquita, Leandro Dragueta Delfino, Victor Spiandor Beretta

**Affiliations:** 1School of Technology and Sciences, São Paulo State University (UNESP), Presidente Prudente 19060-900, SP, Brazil; claudielecarla@gmail.com; 2Laboratory of InVestigation in Exercise—LIVE, Graduate Program in Movement Sciences, Department of Physical Education, School of Technology and Sciences, São Paulo State University (UNESP), Presidente Prudente 19060-900, SP, Brazil; amanda.b.santos@unesp.br (A.B.d.S.); isabella.leoci@unesp.br (I.C.L.); enrique.gervasoni@unesp.br (E.G.L.); ewerton.p.antunes@unesp.br (E.P.A.); wesley.torres@unesp.br (W.T.); eduardo.duarte@unesp.br (E.D.d.L.M.); leandro.dragueta@unesp.br (L.D.D.)

**Keywords:** physical activity, workload, teaching

## Abstract

Chronic high stress levels related to work impact the quality of life (QoL). Although physical activity improves QoL, it is not clear whether this life study habit could attenuate possible relationships between QoL and stress in public school teachers. The sample for this study was made up of 231 teachers from public schools. QoL was assessed using the Short-Form Health Survey and physical activity via Baecke’s questionnaire. A Likert scale assessed stress level perception. Poisson Regression analyzed the association between stress level and QoL domains adjusted for sex, age, and socioeconomic conditions (model 1). In model 2, physical activity level was inserted in addition to model 1. Seven out of eight domains of QoL, except the domain of pain, were associated with high levels of stress (all *p* < 0.05–model 1). However, in model 2, the associations of the high levels of stress with general health status (*p* = 0.052) and functional capacity (*p* = 0.081) domains of QoL were mitigated. Our results indicated that physical activity mitigated the relationship between higher levels of stress and lower perception of general health status and functional capacity domains in secondary public school teachers.

## 1. Introduction

Teaching is considered one of the most important professions around the world. Besides spending many hours per week in activities in the classroom, it is common for teachers to perform extracurricular activities (e.g., setting up and correcting exams and being school coordinators), which leads to a high workload [1,2,3]. Extracurricular activities are usually carried out during weekends, which decreases the time for recreational activities and family [4]. In addition, teachers report difficulty in decreasing their working hours due to their demands to improve the teaching quality and the expectations of society [5]. It should be highlighted that almost 90% of teachers in public schools reported an increase in overall hours over the past years [1]. Previous studies have demonstrated that a high workload decreases well-being and affects physical and mental health, such as an increase in stress levels [6,7,8].

Stress results from a hard situation that threatens homeostasis, which could cause a state of worry or mental tension. Stress can be understood as a normal reaction of the body when situations of danger or threat are experienced, causing the body to be in a state of alert [9]. The risk of stress and burnout related to work seems to increase according to the number of working hours [8]. Teachers with a high level of workload can present sleep problems and burnout and reported higher stress levels in different countries [10,11,12]. In a nationwide survey in Japan carried out from June to December 2018, it was observed that the high workload in public junior high school teachers was associated with higher levels of stress [13]. Burnout syndrome is generally considered professional exhaustion caused by excessive work demands [14,15,16]. In most cases, all body systems are affected by stress (e.g., cardiovascular, endocrine, nervous, and muscular systems) [14]. Thus, staying in a state of stress for a long time impacts the organism’s homeostasis and leads to several physical and mental illnesses, such as anxiety, depression, cardiovascular diseases, stomach ulcers, and sleep dysregulation, and impacts social interaction [14,15,16], which impact the quality of life (QoL).

Quality of life is the self-perception that human beings have of their own health, considering emotional, physical, and social conditions [17,18]. Socioeconomic, environmental, and sociocultural aspects can influence quality of life [19,20]. A previous study demonstrated quality of life and psychological well-being are impaired in teachers from Brazil [21]. This mentioned study included 517 teachers from public and private schools in Brazil and evidenced that approximately 20% showed poor indices of general quality of life, ~17% poor quality of life in the physical domain, ~18% in the psychological domain, and ~17% in social domain [21]. In addition, the regression analysis indicated that the meaning of life explained 42% of the variability in the general quality of life and 51% and 27% from the psychological and physical domains, respectively [21]. Although physical activity practice can increase the quality of life in different populations [22,23], the lack of time for physical activity due to the work routine may be related to worse mental health and increased stress in this class of professionals [5].

Physical activity is one lifestyle habit that can improve quality of life and stress levels. Physical activity is defined as any body movement produced by the musculoskeletal system where energy expenditure is above resting levels [24]. Lower frequency of physical activity practice increases the risk of having stress levels, which was exacerbated during the COVID-19 pandemic (i.e., due to social distancing and increased workload due to teleworking) [25,26]. Dias et al. (2017) [27] evidenced that ~72% of Brazilian public school teachers reported insufficient physical activity. Also, approximately 26% of the physical education teachers from Brazil reported did not practice physical activity during COVID-19, and 10.3% increased their consumption of alcoholic drinks [27], which may be due to the stressful and challenging nature of teaching work during this period [28]. On the other hand, physical activity practice was associated with a better quality of life for workers at a university [29]. In addition, previous studies demonstrated that physical activity practice positively affects job satisfaction and job stress [30,31]. A previous study by Bogaert et al., (2014) [32] observed that physical activity practice in leisure time was associated with perceived health in teachers. Although interesting, few studies evaluated the possible effects of physical exercise on quality of life, especially considering the stress level in public school teachers. Teachers with chronic musculoskeletal pain involved in six weeks of yoga improved their pain and mental health scores [33].

Among the gaps that the present study is advancing, considering the specificity of each domain of quality of life in relation to self-perception of stress is one of them, as such procedures will help to understand which domains are related and which are more strongly related than the others. It is also noteworthy that the instrument used in this study to assess quality of life is composed of eight different domains, providing larger domains when compared to the WHOQol [34], for example. Furthermore, this appears to be the first study to evaluate the role of physical activity in the relationship between stress and quality of life in public school teachers.

Therefore, the objective of the present study was to investigate the relationship between stress levels and quality of life in public school teachers in a Brazilian city and verify the role of physical activity in this relationship. We expected to observe associations between lower quality of life and high self-perception of stress in teachers. Also, we expected that physical activity levels could mitigate the relationship between lower quality of life domains and high levels of stress.

## 2. Materials and Methods

All procedures of this cross-sectional observational (type of study in which observations are made over a certain period of time, in which relationships between a dependent variable and independent variables are analyzed) study were approved by the research ethics committee at São Paulo State University (CAAE: 72191717.9.0000.5402). The participants were informed about the procedures and objectives of the research, and those who agreed to participate signed the statement of consent before their participation.

### 2.1. Participants

The sample of the present study was composed of 231 teachers from five geographic regions of the city of Presidente Prudente, located in the State of São Paulo (southeast region of Brazil).

We provided a posteriori post hoc power calculation using the formula proposed by Rosner (2011) [35] for power calculation comparing two groups, considering the average quality of life score from all the eight specific domain scores according to stress level (low-level vs. high-level groups). The results are presented below:$$Power=\varphi \;\left(-Z_{1-a/2}+\frac{\Delta }{\sqrt{\sigma_{1}^{2}/\eta 1+\sigma_{2}^{2}/\eta 2}}\right)$$
where (µ1, σ$\frac{2}{1}$) and (µ2, σ$\frac{2}{2}$) are the means and variances of the two respective groups and η_1_, η_2_ are the sample sizes of the two groups, considering an alfa error of 0.05. When we used z-scored values for the calculation (since our variable was skewed), the power analysis resulted in 82.8%, as presented below:
−(1.96) + 0.335/√ (0.770^2^/76) + (0.934^2^/159)= 0.946 critical *Z* value = 0.828 (converted value) = 82.8% power.


The criteria for inclusion in this research were (i) being a working teacher in the state education network in the city of Presidente Prudente and (ii) agreeing to respond to the questionnaire and have their anthropometric measurements evaluated by signing the Free and Informed Consent Form. The following exclusion criteria were considered: (i) not completely answering any of the study questionnaires and (ii) missing or refusing to take anthropometric measurements.

### 2.2. Data Collection and Analysis

Data collection was carried out in the school environment at a time agreed with the principals and teachers at public schools so as not to disrupt the work routine. All assessments carried out via questionnaire were face-to-face.

#### 2.2.1. Stress Level

The stress was assessed by the perception of the current level of stress. A Likert scale (i.e., very low, low, moderate, intense, and very intense) was used for the participant’s answers. In general, Likert scales can be used in the literature to assess participants’ feelings and perceptions regarding certain topics [36,37]. Thus, this instrument is commonly used to assess stress levels worldwide [38,39]. Based on the responses, those participants who reported very low or low stress were categorized as low stress, while those who reported moderate, intense, or very intense were classified as having a high level of stress.

#### 2.2.2. Quality of Life

Quality of life was assessed using the instrument proposed by Ware et al., denominated Short-Form Health Survey (SF-36), which evaluates eight different domains [40] and had its reproducibility and validity tested in the Brazilian population [41]. The quality of life domains assessed by this instrument are functional capacity, physical limitations, body pain, general health status, vitality, social aspects, emotional aspects, and mental health.

#### 2.2.3. Habitual Physical Activity

Habitual physical activity was assessed by the questionnaire of Baecke et al. [42], which was validated against gold standard methods for measuring physical activity, such as accelerometry, in the Brazilian population [43]. This instrument assesses the different domains of physical activity, considering occupational activities (load you have to lift at work; time you need to walk at work; how much you sweat and feel fatigued), leisure-time sports activities (amount of time, weekly frequency, and intensity of sports practice) and commuting activities (time spent actively commuting to work, shopping, etc…). The sum of the scores of all domains provides a dimensionless score, where the highest value indicates that the participant is more physically active. Because it is an instrument that broadly assesses physical activity in different domains and can also be translated for the Brazilian population, this questionnaire was chosen for the present study.

### 2.3. Statistical Analysis

Data normality was analyzed using the Shapiro–Wilk test. When normality was detected, Student’s *t*-test was performed for independent samples (Low stress vs. High stress), and data were presented as mean and standard deviation. When normality was not detected, the Mann–Whitney test was used, and the variables were presented as median and interquartile range. The association between stress level and quality of life domains was analyzed in two multivariate models: model 1 (adjusted for sex, age, and socioeconomic status) and model 2 (model 1 + physical activity level). To perform this association, a Generalized Linear Model with log-transformed values of domain-specific quality of life scores with robust variance adjustment was used, presenting beta coefficients and 95% confidence intervals considering the stress level as a factor (low level vs. high level). A quantile regression model was used to analyze the association between median (50th percentile) scores of quality of life and stress level, considering the very low stress level as the reference category. The statistical package used was IBM SPSS 25.0, and the statistical significance adopted was 5%.

## 3. Results

The self-reported prevalence of stress by public school teachers can be considered high, as approximately 50% of teachers reported having moderate stress, while 16.8% reported having intense stress. Only 2.8% report experiencing very intense stress. This information is presented in Figure 1.

In Table 1, the variables characterizing the agreement sample are compared with self-reported stress levels (low stress versus high stress). The highest quality of life scores were obtained in the physical limitation and emotional limitation domains in teachers with low stress. There was a significant difference in the domains of physical limitation, general health status, vitality, social aspects, mental health, pain, functional capacity, emotional limitations, all these values being higher in teachers with a low level of stress.

Table 2 presents the multivariate models considering the association between stress level and quality of life in teachers. The domain of physical limitation (*p* < 0.001), general health status (*p* = 0.002), vitality (*p* < 0.001), social aspects (*p* < 0.001), mental health (*p* < 0.001), functional capacity (*p* = 0.006), and emotional limitations (*p* < 0.001) were associated with high levels of stress in model 1 (adjusted for sex, age, and socioeconomic status). Thus, considering model 1, only the domain of pain (*p* = 0.076) was not associated with high levels of stress (Table 2).

The quantile regression model between quality of life and stress level is presented in Table 3. Participants with high stress levels showed lower (poor) median scores of physical limitation and general health status when compared to those with very low stress levels. The median domain scores of vitality, social aspects, and mental health showed a significant decrease (worsening) from moderate to high/very high stress levels when compared to participants with very low stress levels.

## 4. Discussion

Our study aimed to analyze whether the level of perceived stress was related to a worse score of quality of life in teachers considering the level of habitual physical activity. As expected, our first hypothesis was confirmed, as high stress level was associated with a worse score of quality of life in different domains (Table 2). Also, our second hypothesis was partially confirmed. Physical activity practice was able to attenuate the association between worse scores of quality of life and high levels of stress. However, this attenuation was observed in only one (functional capacity) out of seven domains of quality of life. In addition, the prevalence of perceived stress in public school teachers in a Brazilian city can be considered high.

Teachers with a high level of perceived stress have lower scores in seven of the eight domains of quality of life when compared to teachers with a low level of stress. Furthermore, the high level of stress was inversely related to better quality of life. Our study evidenced that stress level was associated with mental health and emotional role domains. A possible influence, at least in part, is the fact that high levels of stress are related to burnout syndrome, as demonstrated by Australian teachers [44]. Burnout syndrome and stress can lead to mental health issues (e.g., anxiety and depression) [45], which could directly impact the quality of life domains such as mental health and emotional role. Also, our results indicate an inverse relationship between stress and quality of life domains such as general state of health and vitality, which may be influenced by fatigue [46]. The fatigue arising from stress seems to be a key factor in reducing the perception of health and vitality [46]. High levels of stress were also inversely related to functional capacity. One of the mechanisms that may act in this relationship is that stress would contribute to increases in cortisol levels [47]. Elevated cortisol levels for a long period could impair muscle function, which could contribute to a decrease in functional capacity [48].

Despite this association between stress level and quality of life, physical activity has been considered an important tool in reducing stress and increasing the quality of life of teachers [49,50]. In the present study, the associations between high levels of stress and functional capacity were attenuated after the inclusion of physical activity in the adjusted model. Physical activity practice reduces the cortisol level and contributes to the increase in hormones related to well-being feeling, such as endorphins [51]. Thus, these effects of exercise on hormonal function could explain, at least in part, the attenuation of the association between stress and quality of life. Demmin et al. [49], in a study with American teachers, observed that meditation linked to aerobic exercise contributed to mental health and well-being in teachers during the COVID-19 pandemic, which would certainly contribute to a better perception of quality of life.

Regarding the attenuation of the association between stress and functional capacity, it is well-established that physical activity contributes to an increase in muscle strength and muscle activity [52]. The constant practice of physical activity, mainly via strength and multimodal exercises, has been associated with improvements in functional capacities such as strength, coordination, agility, and balance [53,54]. Thus, these effects of exercise on functional capacity domains could explain, at least in part, the attenuation of the association between stress and quality of life demonstrated in our study.

It should be highlighted in our study that we observed that approximately 70% of school teachers reported being stressed. Similar results were evidenced in previous studies [55,56] with different nationalities. Parthasarathy et al. observed that 85% of American school teachers reported being sometimes/often stressed [55]. Biernat et al. [56] also demonstrated high values in the prevalence of self-reported stress in teachers from Poland. The high workload, insufficient remuneration, and poor infrastructure conditions are some of the reasons reported by teachers that may contribute to this condition of high perceived stress level.

Although presenting interesting and relevant findings, our study also presents limitations that should be considered. The present study has a cross-sectional design, which does not allow us to consider causal relationships. Also, we determined the habitual physical activity practice via a questionnaire, which makes it difficult to determine the intensities of physical activity as assessed by accelerometry. The self-report physical activity practice by questionnaire is susceptible to the bias of memory and should be considered in the interpretation of the results. However, it should be highlighted that the Baecke questionnaire is validated for the Brazilian population when compared to gold standard methods for measuring physical activity (i.e., accelerometry). In addition, the attenuation of the relationship between higher levels of stress and quality of life domains in public school teachers in a Brazilian city should be considered with caution. To confirm this attenuation and to what extent physical activity acts as a mediator or moderator in the relationship, at least in parts, a mediation analysis would be more appropriate. However, our data did not meet the basic assumptions, and therefore, we performed the statistical analysis using Poisson Regression, with the covariates being inserted in a hierarchical model. Another limitation to be reported is the absence of a focus group in the present study or the conduct of interviews with teachers, which could help to better understand the possible causes of stress and also the self-perception of these professionals regarding this situation.

One of the novelties of the present study was to consider the analysis of the specificity of each quality of life domain separately. Another point is that this appears to be the first study to investigate the role of physical activity in the relationship between perceived stress and quality of life in teachers, with attenuation being observed in only one of the domains (functional capacity) in this important professional class.

As strong points, we emphasize the fact that this study was carried out in a developing country like Brazil, presenting the characteristics of countries in this economic condition since most of the data have been from developed countries. As practical applications, we emphasize the importance of encouraging physical activity in teachers in order to mitigate, at least in some domains, the relationship between high levels of stress and lower quality of life.

As suggestions for future studies, longitudinally analyzing the cause-and-effect associations between stress and QoL and the role of physical activity is something that needs to be advanced. Considering strategies for reducing stress in teachers, investigating the effectiveness of intervention programs via physical activity in the school environment with teachers could be a good alternative to reduce the stress load on these professionals.

## 5. Conclusions

Approximately 70% of teachers participating in this study reported having high levels of stress. Higher levels of stress were associated with seven out of eight domains of the quality of life. However, the physical activity practice attenuated the relationship between higher levels of stress and functional capacity domains of quality of life in secondary public school teachers in a Brazilian city. Therefore, encouraging teachers to practice physical activity can be an alternative for reducing stress levels and improving quality of life. Programs that could contribute to increasing physical activity in the workplace, for example, could be an alternative to be tested for these objectives.

## Figures and Tables

**Figure 1 ijerph-21-00088-f001:**
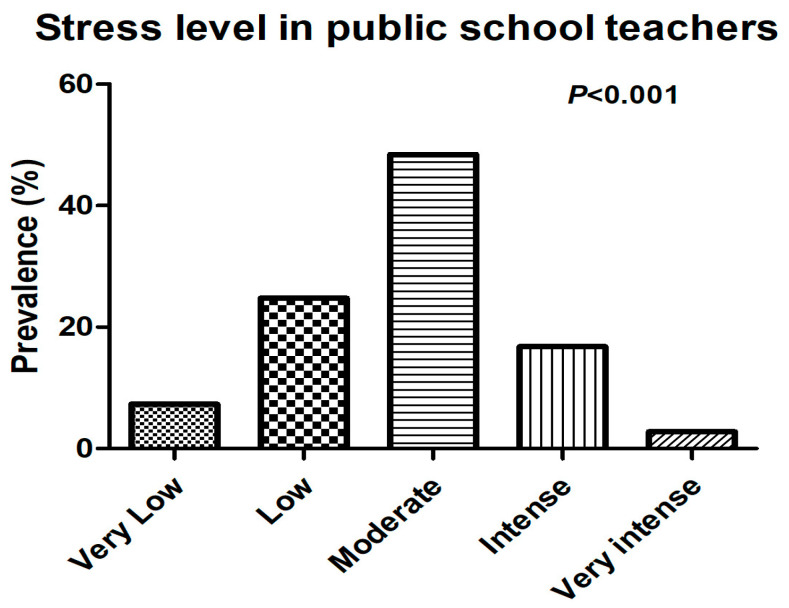
Self-reported prevalence of stress by public school teachers.

**Table 1 ijerph-21-00088-t001:** Sample characteristics.

	Low Stress	High Stress	
	Median (II)	Median (II)	*p*-Value
Age (years) #	46.08 ± 10.24	44.84 ± 10.55	0.396
Physical limitation (0–100)	100.00 (25)	75.00 (75.00)	<0.001 *
General health status (0–100)	77 (28.0)	67.0 (30.0)	0.004 *
Vitality (0–100)	70 (20)	45.0 (25.0)	<0.001 *
Social aspects (0–100)	88.0 (20.0)	63.0 (38.0)	<0.001 *
Mental health (0–100)	80.0 (16.0)	60.0 (28.0)	<0.001 *
Pain (0–100)	44.0 (9.0)	42.0 (9.0)	0.026
Functional capacity (0–100)	90.0 (30.0)	80.0 (40.0)	0.013
Emotional limitations (0–100)	100.0 (10.0)	100.0 (67.0)	<0.001
Physical activity (Baecke’s score)	7.04 (2.79)	7.48 (2.04)	0.201

# Presented as mean and standard deviation, as data normality was detected. * indicates the statistical difference between teachers classified as having low and high stress levels.

**Table 2 ijerph-21-00088-t002:** Association between stress and quality of life in public school teachers (n = 231).

			Physical Limitation	General Health Status	Vitality	Social Aspects	Mental Health	Pain	Functional Capacity	Emotional Limitations
Stress level	Model 1	β	−0.25	−0.13	−0.39	−0.23	−0.31	−0.06	−0.13	−0.25
		95% CI	−0.37; −0.13	−0.22; −0.05	−0.49; −0.29	−0.30; −0.16	−0.40; −0.22	−0.12; 0.01	−0.22; −0.04	−0.37; −0.13
		*p*-value	<0.001	0.003	<0.001	<0.001	<0.001	0.076	0.004	<0.001
	Model 2	β	−0.25	−0.13	−0.39	−0.23	−0.31	−0.06	−0.12	−0.25
		95% CI	−0.37; −0.13	−0.22; −0.05	−0.49; −0.29	−0.30; −0.16	−0.40; −0.22	−0.12; 0.01	−0.21; −0.04	−0.37; −0.13
		*p*-value	<0.001	0.002	<0.001	<0.001	<0.001	0.067	0.006	<0.001

Model 1: Adjusted for sex, age, and socioeconomic status; Model 2: adjusted for model 1 + physical activity.

**Table 3 ijerph-21-00088-t003:** Quantile regression model for the association between stress level and quality of life domain scores in public school teachers (n = 231).

	β Coefficient for 50th Percentile (95% Confidence Interval)							
	Physical Limitation	General Health Status	Vitality	Social Aspects	Mental Health	Pain	Functional Capacity	Emotional Limitations
Stress level								
Very low	Reference	Reference	Reference	Reference	Reference	Reference	Reference	Reference
Low	0.1 (−15.4; 15.4) *p* = 0.999	−6.0 (−19.1; 7.2) *p* = 0.372	−9.1 (−20.5; 2.3) *p* = 0.119	−12.0 (−26.9; 2.9) *p* = 0.114	−4.7 (−14.6; 5.2) *p* = 0.348	−0.1 (−4.9; 4.9) *p* = 0.999	1.9 (−14.6; 18.4) *p* = 0.817	−0.1 (−20.4; 20.4) *p* = 0.999
Moderate	−0.1 (−14.5; 14.5) *p* = 0.999	−12.2 (−24.5; 0.2) *p* = 0.054	**−28.8** **(−39.6; −18.1)** ***p* < 0.001**	**−25.0** **(−38.9; −11.1)** ***p* < 0.001**	**−18.9** **(−28.2; −9.6)** ***p* < 0.001**	−2.0 (−6.7; 2.6) *p* = 0.398	−6.5 (−22.1; 8.9) *p* = 0.406	−0.1 (−19.1; 19.1) *p* = 0.999
High	**−50.0** **(−66.4; −33.6)** ***p* < 0.001**	**−23.4** **(−37.4; −9.4)** ***p* = 0.001**	**−47.9** **(−60.1; −35.8)** ***p* < 0.001**	**−50.0** **(−65.7; −34.3)** ***p* < 0.001**	**−39.1** **(−49.6; −28.6)** ***p* < 0.001**	−2.0 (−7.3; 3.2) *p* = 0.454	−15.3 (−32.8; 2.2) *p* = 0.086	−0.1 (−21.6; 21.6) *p* = 0.999
Very high	−25.0 (−53.8; 3.8) *p* = 0.089	−4.8 (−29.5; 19.8) *p* = 0.699	**−38.2** **(−59.6; −16.8)** ***p* < 0.001**	**−50.0** **(−77.7; −22.3)** ***p* < 0.001**	**−67.3** **(−85.8; −49.0) ** ***p* < 0.001**	−2.0 (−11.3; 7.3) *p* = 0.671	5.9 (−25.0; 36.8) *p* = 0.707	−33.0 (−71.1; 5.1) *p* = 0.089

Analysis adjusted by sex, age, socioeconomic status, and physical activity; Bold represents statistical significance.

## Data Availability

The data presented in this study are available upon request from the corresponding author. The data are not publicly available due to privacy.

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
