# Peer review of "The Association between Perceived Stress, Quality of Life, and Level of Physical Activity in Public School Teachers"

_ijerph, 2024, doi:10.3390/ijerph21010088_

Round 1

Reviewer 1 Report (Previous Reviewer 2)

Comments and Suggestions for Authors

I would like to thank the authors for their answers. The changes introduced in the study enrich it and make it easier to read. However, I believe that the summary should be expanded.

Author Response

Thank you for your previous comments and suggestions, which have helped us to improve the quality of the manuscript. 

Reviewer 2 Report (Previous Reviewer 3)

Comments and Suggestions for Authors

I believe that the authors have made an important and positive effort to respond to the guidelines proposed by the reviewers, improving the article with respect to its initial version.  Although there are aspects that could be improved, the authors justify it conveniently in the limitations, so in my view, it meets the minimum requirements to be published.

Best regards.

Author Response

Thank you for your previous comments and suggestions, which have helped us to improve the quality of the manuscript. 

Reviewer 3 Report (Previous Reviewer 4)

Comments and Suggestions for Authors

Author Response

Reviewer comment:

Thank you for your response and for taking into consideration the feedback provided. I understand the significance of exploring the relationship between physical activity and the well-being of teaching professionals. However, I would like to point out that the scientific literature extensively covers this topic, with numerous studies already investigating the implications of working conditions and workload in the educational setting. Considering the substantial body of prior research, it is crucial that your study contributes meaningful and novel insights to the field. I recommend a thorough review of existing literature to identify specific areas where your research can stand out for its originality or offer fresh perspectives.

Response: Dear reviewer, thank you for your comment. We agree that there are numerous studies already investigating the implications of working conditions and workload in the educational environment. However, we are advancing in some aspects, the first is to consider the specificities of different domains, considering the assessment by the SF-36 instead of the WHOqOL for example, where 8 domains are assessed instead of 4. Another point is that it is not well established whether the practice of physical activity could play a role in this relationship, especially considering this class of professionals. Therefore, based on this reviewer's comments, we have inserted the following paragraph into the introduction and we put these aspects into the discussion as well, showing what’s new in the study and offering new perspectives.

Introduction

“Among the gaps that the present study is advancing, considering the specificity of each domain of quality of life in relation to self-perception of stress is one of them, as such procedures will help to understand which domains are related and which are more strongly related than the others. It is also noteworthy that the instrument used in this study to assess quality of life is composed of eight different domains, providing larger domains when compared to the WHOQol [35], for example. Furthermore, this appears to be the first study to evaluate the role of physical activity in the relationship between stress and quality of life in public school teachers.”

Discussion

“One of the novelties of the present study was to consider the analysis of the speci-ficity of each quality of life domain separately. Another point is that this appears to be the first study to investigate the role of physical activity in the relationship between perceived stress and quality of life in teachers, with attenuation being observed in only one of the domains (functional capacity) in this important professional class.”.

Reference:

[35] Harper, A.; Power, M.; Orley, J.; Herrman, H.; Schofield, H.; Murphy, B.; Metelko, Z.; Szabo, S.; Piber-nik-Okanovic, M.; Quemada, N.; et al. Development of the World Health Organization WHOQOL-BREF Quality of Life Assessment. The WHOQOL Group. Psychol Med 1998, 28, 551–558, doi:10.1017/S0033291798006667.

Reviewer comment:

Thank you for providing clarification on your sample calculation. However, I would like to point out that there appears to be a potential confusion between a priori and a posteriori sample size calculations in your response. In your explanation, you mention considering an alpha error of 5% and a confidence interval of 95%, requiring 230 participants for 80% power. It is essential to distinguish between a priori sample size calculations, which are determined before data collection, and a posteriori sample size calculation, which are conducted after data analysis. It is unclear from your response whether the power calculation is based on the actual analyses performed in your study or if it was predetermined before data collection. To ensure a comprehensive understanding of the study's statistical robustness, it would be beneficial if you could provide details on the power of the analyses conducted in your work. This information will contribute to a more accurate assessment of the reliability of your findings.

Response: Dear reviewer, the sample size calculation was made a priori of data collection. Aiming to address your concern, we provided a posteriori post-hoc power calculation using the formula proposed by Rosner (2011) for power calculation comparing two groups, considering the average quality of life score from all the eight specific domain-scores according to stress level (low level vs. high level groups). The result is presented below:

where (µ1, σ), (µ2, σ) are the means and variances of the two respective groups, and n1, n2 are the sample sizes of the two groups, considering an alfa error of 0.05.

Our results were:

- (1.96) + 14.44 / √ (12.99²/76) + (15.77²/159)

= 5.463 critical Z value = 1 (converted value) = 100% power

When we used z-scored values for calculation (since our variable was skewed), the power analysis resulted in 82.8%, as presented below:

- (1.96) + 0.335 / √ (0.770²/76) + (0.934²/159)

= 0.946 critical Z value = 0.828 (converted value) = 82.8% power

Reference:

Rosner B. Chapter 8 – Hypothesis Testing: two-sample inference. 8.10 Estimation of Sample Size and Power for Comparing Two Means, p.301. Fundamentals of Biostatistics. 7th ed. Boston, MA: Brooks/Cole; 2011.

This information was entered in the methods section:

“The sample of the present study was composed of 231 teachers from five geo-graphic regions of the city of Presidente Prudente located in the State of São Paulo (southeast region of Brazil).

We provided a posteriori post-hoc power calculation using the formula proposed by Rosner (2011) [36] for power calculation comparing two groups, considering the average quality of life score from all the eight specific domain scores according to stress level (low-level vs. high-level groups). The result is presented below:

where (µ1, σ), (µ2, σ) are the means and variances of the two respective groups, and n1, n2 are the sample sizes of the two groups, considering an alfa error of 0.05. When we used z-scored values for calculation (since our variable was skewed), the power analysis resulted in 82.8%, as presented below:

- (1.96) + 0.335 / √ (0.770²/76) + (0.934²/159)

= 0.946 critical Z value = 0.828 (converted value) = 82.8% power.”.

Reviewer comment:

Thank you for addressing my comments. I appreciate your efforts in analyzing the distribution of data and respecting the assumptions, particularly the consideration of nonparametric characteristics for most variables. However, I would like to express concerns regarding the choice of Poisson Regression for analyzing associations, especially if the dependent variables are not counts. Poisson Regression is typically suitable for count data, and its use may not be the most appropriate when dealing with non-count variables. To ensure the validity of your analyses, I recommend considering alternative non-parametric regression models that are more suitable for non-count data. Some examples include: a) Logistic Regression: Appropriate for binary or dichotomous dependent variables. b) Quantile Regression: Useful when analyzing the effect of predictors on different quantiles of the dependent variable distribution. I suggest reevaluating your statistical approach and selecting a regression model that aligns better with the nature of your dependent variables. This adjustment will enhance the robustness and validity of your findings.

Response: Dear reviewer, we appreciate the insightful comment. Once our main objective was to investigate whether participants with high stress levels will be associated with poor quality of life domain scores when compared to those with low stress levels, we adopted a Generalized Linear Model analysis by defining stress level as a factor (high level vs. low level) for prediction of log-transformed quality of life domain scores. The significant associations remained in the same direction and our results shown in Table 2 were updated for this new analysis approach as well as the statistical analysis description, please see below:

“To perform this association, a Generalized Linear Model with log-transformed values of domain-specific quality of life scores with robust variance adjustment was used, presenting beta coefficients and 95% confidence intervals considering the stress level as a factor (low level vs. high level).”.

Table 2. Association between stress and quality of life in public school teachers (n = 231).

Model 1

Model 2

β

95%CI

P-value

β

95%CI

P-value

Physical limitation

-0.25

-0.37; -0.13

<0.001

-0.25

-0.37; -0.13

<0.001

General health status

-0.13

-0.22; -0.05

0.003

-0.13

-0;22; -0.05

0.002

Vitality

-0.39

-0.49; -0.29

<0.001

-0.39

-0.49; -0.29

<0.001

Social aspects

-0.23

-0.30; -0.16

<0.001

-0.23

-0.30; -0.16

<0.001

Mental health

-0.31

-0.40; -0.22

<0.001

-0.31

-0.40; -0.22

<0.001

Pain

-0.06

-0.12; 0.01

0.076

-0.06

-0.12; 0.01

0.067

Functional capacity

-0.13

-0.22; -0.04

0.004

-0.12

-0.21; -0.04

0.006

Emotional limitations

-0.25

-0.37; -0.13

<0.001

-0.25

-0.37; -0.13

<0.001

Model 1: Adjusted for sex, age, and socioeconomic status; Model 2: Adjusted for model 1+ Physical activity.

Reviewer comment:

If the authors dichotomized the dependent variable (I want to assume this from a comment that appears in the discussion), I have some comments: - Where is the way in which the categorization of the dependent variable was done? This should obligatorily appear in the statistical analysis section. - Why did the authors report the beta coefficients and not as odds ratios? This does not help the interpretation by the readers.

Response: Dear reviewer, we apologize for the sentences that appear in some dichotomization of the outcome, so we carefully revised the Discussion section to fit with our analysis. Our outcome was the domain scores of quality of life in continuous form and, for this reason, it was reported beta coefficient according to low vs. high-stress level (main predictor variable as factor).

Round 2

Reviewer 3 Report (Previous Reviewer 4)

Comments and Suggestions for Authors

Thank you for inviting me again to review this manuscript. The authors have more or less resolved some of the comments made. However, there is one point that they have not yet resolved. 

The problem is that Table 2 is inverted. Normally, the left column corresponds to predictors (and not outcomes). The outcomes should appear on the right side of the table (hence the confusion generated). It would have helped, if this approach had been chosen, to indicate in the table footnote that the reference is "low stress".

However, the authors still do not justify why they dichotomize the variable "stress" into low versus high. Did they use validated cut-off points? Why do they not use the original Likert scale? This would certainly provide a better understanding of the relationship between the variables. Also, if the outcome is continuous, have the authors checked the assumptions of normality of the residuals? Linearity? Heteroscedasticity? Etc.? I would really like to see if the relationship is linear or curvilinear. Having a certain level of stress (low or moderate) could be associated with a "better" quality of life compared to a very low level of stress. 

"Based on the responses, those participants who reported very low or low stress were categorized as having low stress, while those who reported moderate, intense, or very intense stress were classified as having a high level of stress".

Why this decision? There is no justification. They have not referenced any papers that adopt this methodology. 

Best wishes,

Author Response

Response:

Dear reviewer, 

We thank for the insightful comments once again. 

- The Table 2 was adjusted accordingly. 

- Regarding the dichotomization of the stress level, this procedure was adopted to characterize the sample according to stress level (low vs. high), as similarly made in a previous study (Ganesh et al., 2018). 

Reference: 

Ganesh R, Mahapatra S, Fuehrer DL, et al. The Stressed Executive: Sources and Predictors of Stress Among Participants in an Executive Health Program. Glob Adv Health Med. 2018;7:2164956118806150. Published 2018 Oct 17. doi:10.1177/2164956118806150 

- About the linear regression analysis in Table 2, we run the regression models using log-transformed values due to the initial skewed distribution of data. We checked the assumptions for linear regression analysis, which is presented in the frame below: 

Predictor:  

Low vs. high stress 

Independence of residuals 

Collinearity 

Homocedasticity 

Outcome variable 

Durbin-Watson 

VIF 

ZRESID vs ZPRED 

Physical limitation 

1.795 

1.030 

Scatterplot without tendency (homoscedastic) 

General health status 

1.658 

1.027 

Scatterplot without tendency (homoscedastic) 

Vitality 

1.623 

1.034 

Scatterplot without tendency (homoscedastic) 

Social aspects 

1.680 

1.033 

Scatterplot without tendency (homoscedastic) 

Mental health 

1.637 

1.027 

Scatterplot without tendency (homoscedastic) 

Pain 

2.069 

1.030 

Scatterplot without tendency (homoscedastic) 

Functional capacity 

1.613 

1.020 

Scatterplot without tendency (homoscedastic) 

Emotional limitations 

1.722 

1.029 

Scatterplot without tendency (homoscedastic) 

- Aiming to address your request, we run a quantile regression analysis (Table 3), by considering the median (50th percentile) values of quality of life domain-scores according to stress level, where very low stress level was the reference category. This analysis allowed us to verify whether median QoL domain-scores were different according to each level of stress. 

Table 3. Quantile regression model for association between stress level and quality of life domain-scores in public school teachers (n = 231). 

β coefficient for 50th percentile (95% confidence interval) 

Physical limitation 

General health status 

Vitality 

Social aspects 

Mental health 

Pain  

Functional capacity 

Emotional limitations 

Stress level 

Very low 

Reference 

Reference 

Reference 

Reference 

Reference 

Reference 

Reference 

Reference 

Low 

0.1  

(-15.4; 15.4) p=0.999 

-6.0 

(-19.1; 7.2) p=0.372 

-9.1 

(-20.5; 2.3) p=0.119 

-12.0 

(-26.9; 2.9) p=0.114 

-4.7 

(-14.6; 5.2) p=0.348 

-0.1 

(-4.9; 4.9) p=0.999 

1.9 

(-14.6; 18.4) p=0.817 

-0.1 

(-20.4; 20.4) p=0.999 

Moderate 

-0.1 

(-14.5; 14.5) p=0.999 

-12.2 

(-24.5; 0.2) p=0.054 

-28.8 

(-39.6; -18.1) p<0.001 

-25.0 

(-38.9; -11.1) p<0.001 

-18.9 

(-28.2; -9.6) p<0.001 

-2.0 

(-6.7; 2.6) p=0.398 

-6.5 

(-22.1; 8.9) p=0.406 

-0.1 

(-19.1; 19.1) p=0.999 

High 

-50.0  

(-66.4; -33.6) p<0.001 

-23.4 

(-37.4; -9.4) p=0.001 

-47.9 

(-60.1; -35.8) p<0.001 

-50.0 

(-65.7; -34.3) p<0.001 

-39.1 

(-49.6; -28.6) p<0.001 

-2.0 

(-7.3; 3.2) p=0.454 

-15.3 

(-32.8; 2.2) p=0.086 

-0.1 

(-21.6; 21.6) p=0.999 

Very high 

-25.0  

(-53.8; 3.8) p=0.089 

-4.8 

(-29.5; 19.8) p=0.699 

-38.2 

(-59.6; -16.8) p<0.001 

-50.0 

(-77.7; -22.3) p<0.001 

-67.3 

(-85.8; -49.0) p<0.001 

-2.0 

(-11.3; 7.3) p=0.671 

5.9 

(-25.0; 36.8) p=0.707 

-33.0 

(-71.1; 5.1) p=0.089 

Analysis adjusted by sex, age, socioeconomic status, and physical activity. 

This manuscript is a resubmission of an earlier submission. The following is a list of the peer review reports and author responses from that submission.

Round 1

Reviewer 1 Report

Comments and Suggestions for Authors

 Dear Author

The Article “Association between perceived stress, quality of life and level of physical activity in public school teachers, is well organized, with scientific rigor and is of relevance to the scientific community in general and particularly to the teaching community. However, they must be considered small errors that must be considered and corrected so that it is ready to be published.

1-Materials and Methods:

Refers to the type of study but does not justify why it is a cross-sectional observational study, it must justify the type of study

The sample inclusion and exclusion criteria are not presented. They must present

They do not mention what type of statistical tests they used. They must mention and justify

2-Discussion, line 172, states that “Also, our second hypothesis was partially confirmed”, however there is no reference to the first Hypothesis. They must correct

with my regards

Author Response

We would like to thank the editor for the opportunity to resubmit the attached manuscript entitled “Association between perceived stress, quality of life, and level of physical activity in public school teachers” for possible publication in the International Journal of Environmental Research and Public Health.

The authors also wish to express their gratitude or the efforts of the editor and reviewers in directing the manuscript toward a more acceptable form for publication. The manuscript has been carefully checked, and appropriate changes have been made in accordance with the reviewers’ suggestions.

The authors hope that the added revisions adequately address the comments.

Reviewer 1

The Article “Association between perceived stress, quality of life and level of physical activity in public school teachers, is well organized, with scientific rigor and is of relevance to the scientific community in general and particularly to the teaching community. However, they must be considered small errors that must be considered and corrected so that it is ready to be published.

Response: We thank the reviewer for the comment, and we tried to implement all the suggestions in the manuscript.

1-Materials and Methods:

Refers to the type of study but does not justify why it is a cross-sectional observational study, it must justify the type of study

Response: Dear reviewer, as suggested, we inserted this information in the methods section. However, it can also be viewed below:

“All procedures of this cross-sectional observational (type of study in which observations are made over a certain period of time, in which relationships between a dependent variable and independent variables are analyzed) study were approved by the research ethics committee at São Paulo State University (CAAE: 72191717.9.0000.5402)”.

The sample inclusion and exclusion criteria are not presented. They must present

Response: Dear reviewer, as suggested, we inserted the following sentences regarding the inclusion and exclusion criteria.

The criteria for inclusion in this research were: i) being a working teacher in the state education network in the city of Presidente Prudente; ii) agreeing to respond to the questionnaire and have their anthropometric measurements evaluated, by signing the Free and Informed Consent Form. The following exclusion criteria were considered: i) not completely answering any of the study questionnaires; ii) missing or refusing to take anthropometric measurements.

They do not mention what type of statistical tests they used. They must mention and justify

Response: Dear reviewer, the description of the statistical analysis is below and can also be viewed in the methods section:

“Data normality was analyzed using the Shapiro-Wilks test. When normality was detected, the student’s t test was performed for independent samples (Low stress vs High stress) and data were presented as mean and standard deviation. When normality was not detected, the Mann-Whitney test was used, and the variables were presented as median and interquartile range. The association between stress level and quality of life domains was analyzed in two multivariate models: model 1 (adjusted for sex, age, and socioeconomic status) and model 2 (model 1 + physical activity level). To perform this association, Poisson Regression with robust variance adjustment was used. The statistical package used was IBM SPSS 25.0 and the statistical significance adopted was 5%”.

2-Discussion, line 172, states that “Also, our second hypothesis was partially confirmed”, however there is no reference to the first Hypothesis. They must correct with my regards

Response: Dear reviewer, as suggested, such information was included in the first paragraph of the discussion.

“Our study aimed to analyze whether the level of perceived stress was related to a worse quality of life in teachers considering the level of habitual physical activity. As expected, our first hypothesis was confirmed, as high stress level was associated with worse QoL in different domains (Table 2). Also, our second hypothesis was partially confirmed.”

Reviewer 2 Report

Comments and Suggestions for Authors

The article concerns an important issue: teachers' stress and its impact on the quality of life.

My suggestions for development:

Abstract

To make reading the study easier, the Authors should use full names rather than abbreviations in the description. It would be worth specifying which group of teachers the research concerned, i.e., teachers of primary and secondary schools. The only information that appeared in the study refers to public institutions.

Introduction

In the study, it would be worth explaining what the authors mean by stress, as it was described in the case of burnout syndrome. Explaining this concept and its characteristics will make reading the study much easier.

Have the authors considered the possibility of expanding the literature review to include similar research in other countries? This would significantly enrich the presentation of the essence of the problem addressed by the Authors in this study. It would be worth pointing out a broader approach to the topic, including research in other countries.

- lines 61-62 – would the authors be able to indicate which group were teachers of private schools and which of public schools? – this is due to the fact that in the study the authors referred only to public school teachers. It is also worth considering whether there are differences in the way teachers feel stress in public and private schools.

Materials and methods

The discussion of the chapter presented is very general, it is not known what the research questions were and what answers were obtained to individual questions. There is no information on whether the same number of respondents answered each question. It requires expansion to make it possible to refer to the data presented in Tables 1 and 2.

The Authors use the phrases low and high stress levels. It would be worth defining what they mean by these terms.

The study lacks justification as to why the study used, for example, the SF-36 questionnaire or the questionnaire proposed by Baecke.

Discussion

The Authors point to research hypotheses that have not been formulated before. They should be formulated clearly, e.g., in the materials and methods chapter. It would also be worth defining directions for further research as well as ways to reduce stress among the studied professional group.

Conclusions

It would be worth expanding this chapter and also presenting quantitative data.

Author Response

Reviewer 2

The article concerns an important issue: teachers' stress and its impact on the quality of life.

My suggestions for development:

Abstract

To make reading the study easier, the Authors should use full names rather than abbreviations in the description. It would be worth specifying which group of teachers the research concerned, i.e., teachers of primary and secondary schools. The only information that appeared in the study refers to public institutions.

Response: Dear reviewer, as suggested, we reduced the number of abbreviations and included more information about the teachers, as secondary school teachers were evaluated in our study.

Introduction

In the study, it would be worth explaining what the authors mean by stress, as it was described in the case of burnout syndrome. Explaining this concept and its characteristics will make reading the study much easier.

Response: Dear reviewer, thank you for your comment. We added an explanation of the concept of stress in the introduction in accordance with the World Health Organization.

“Stress can be understood as a normal reaction of the body when situations of danger or threat are experienced, causing the body to be in a state of alert [10].”

Have the authors considered the possibility of expanding the literature review to include similar research in other countries? This would significantly enrich the presentation of the essence of the problem addressed by the Authors in this study. It would be worth pointing out a broader approach to the topic, including research in other countries.

Response: Dear reviewer, thank you for your comment. Based on your comment, we added studies from different countries to better contextualize this problem in the introduction.

“The risk of stress and burnout related to work seems to increase according to the number of working hours [9]. Teachers with a high level of workload can present sleep problems, burnout, and reported more stress levels in different countries [11–13]. A study carried out with 635 teachers from 47 Americans from September to October 2020 showed that stress levels were high and more prevalent in female teachers [14]. In a nationwide survey in Japan carried out from June to December 2018, it was observed that the high workload in public junior high school teachers was associated with higher levels of stress [15].”.

- lines 61-62 – would the authors be able to indicate which group were teachers of private schools and which of public schools? – this is due to the fact that in the study the authors referred only to public school teachers. It is also worth considering whether there are differences in the way teachers feel stress in public and private schools.

Response: Dear reviewer, in this case, unfortunately, it would not be possible, as the aforementioned article does not provide standardized analyzes by type of school in its results.

Materials and methods

The discussion of the chapter presented is very general, it is not known what the research questions were and what answers were obtained to individual questions. There is no information on whether the same number of respondents answered each question. It requires expansion to make it possible to refer to the data presented in Tables 1 and 2.

Response: Dear reviewer, 231 teachers from secondary schools in a Brazilian city in the southeast region of Brazil participated in the present study. All participants in this study, who were included in the analysis, answered all questions on the instruments applied. This information was included in our methods.

The Authors use the phrases low and high stress levels. It would be worth defining what they mean by these terms.

Response: Dear reviewer, we use this dichotomization to make the analysis clearer for readers. However, we highlight and try to clarify our methods so that the terms are clearer.

“The stress was assessed by the perception of the current level of stress. A Likert scale (i.e., very low, low, moderate, intense, and very intense) was used for the participant’s answers. In general, Likert scales can be used in the literature to assess participants' feelings and perceptions regarding certain topics [39–41]. Thus, this instrument is commonly used to assess stress levels worldwide [42,43]. Based on the responses, those participants who reported very low or low stress were categorized as low stress, while those who reported moderate, intense, or very intense were classified as having a high level of stress.”.

The study lacks justification as to why the study used, for example, the SF-36 questionnaire or the questionnaire proposed by Baecke.

Response: Dear reviewer, the SF-36 was chosen to assess the quality of life because it is one of the most used instruments in the world for this purpose because it encompasses different domains of quality of life, and because it is validated for the Brazilian population.

The questionnaire by Baecke et al. It is widely used in epidemiological studies involving different populations and analyzing different domains of physical activity (occupational physical activity, leisure-time sports, and commuting activities). This instrument is also validated for the Brazilian population. This information was inserted into the methods.

“Quality of life was assessed using the SF-36 questionnaire proposed by Ware et al. [44] and validated for the Brazilian population [45]. This questionnaire has eight quality-of-life domains: functional capacity, physical limitations, body pain, general health status, vitality, social aspects, emotional aspects, and mental health. The score of this instrument ranges from 0-100, where the highest score indicates a better quality of life. In the present study, quality of life was divided into quartiles, and participants located in the highest quartile (Q4) were considered to have a high quality of life, while participants in the lower quartiles were considered to have a low quality of life. This instrument was chosen to be used in the present study due to the different QoL domains it assesses and because it is validated for the Brazilian population.”

“Habitual physical activity was assessed by the questionnaire of Baecke et al., [46] which was validated against gold standard methods for measuring physical activity, such as accelerometry, in the Brazilian population [47]. This instrument assesses the different domains of physical activity, considering occupational activities (load you have to lift at work; time you need to walk at work; how much you sweat and feel fatigued), leisure-time sports activities (amount of time, frequency weekly and intensity of sports practice) and commuting activities (time spent actively commuting to work, shopping, etc...). The sum of the scores of all domains provides a dimensionless score, where the highest value indicates that the participant is more physically active. Because it is an instrument that broadly assesses physical activity in different domains and can also be translated for the Brazilian population, this questionnaire was chosen for the present study.”.

Discussion

The Authors point to research hypotheses that have not been formulated before. They should be formulated clearly, e.g., in the materials and methods chapter. It would also be worth defining directions for further research as well as ways to reduce stress among the studied professional group.

Response: Dear reviewer, we thank you for your comment and have inserted the suggestion at the end of the discussion section.

“As suggestions for future studies, longitudinally analyzing the cause and effect associations between stress and QoL and the role of physical activity is something that needs to be advanced. Considering strategies for reducing stress in teachers, investigating the effectiveness of intervention programs through physical activity in the school environment with teachers could be a good alternative to reduce the stress load on these professionals”.

Conclusions

It would be worth expanding this chapter and also presenting quantitative data.

Response: Dear reviewer, based on your comments and those of other reviewers, we have included quantitative information about the prevalence of stress levels in teachers, as well as strategies that can contribute to combating this problem in this class of teaching professionals.

“Approximately 70% of teachers participating in this study reported having high levels of stress. Higher levels of stress were associated with seven out of eight domains of the quality of life. However, the physical activity practice attenuated the relationship between higher levels of stress and lower perception of general health status and functional capacity domains of quality of life in secondary public school teachers in a Brazilian city. Therefore, encouraging teachers to practice physical activity can be an alternative for reducing stress levels and improving quality of life. Programs that could contribute to increasing physical activity in the workplace, for example, could be an alternative to be tested for these objectives”.

Reviewer 3 Report

Comments and Suggestions for Authors

I believe that the work presented is interesting although it has been extensively researched and has reached the expected conclusions.

The introduction should clarify the different concepts presented, defining them to frame the research.  In this sense, stress and physical activity are mentioned, but they are not defined or framed for a clear understanding.  On the other hand, concepts appear that if they are not developed do not have a clear coherence in the discourse, as is the case of professional burnout.  It would also be interesting, from my point of view, to link the introduction with concepts such as subjective wellbeing.

As for the instruments used, more information should be given on their validity and reliability.  It would have been appropriate to use a mixed methodology to deepen the knowledge of the subject, using focus groups and interviews, as they have proven to be very valid instruments to investigate people's perceptions.

With respect to the results, I consider that they are unbalanced in relation to the rest of the sections.  They should be developed further.

The limitations should appear in a clearer and more developed way.  It should be borne in mind that in no case can we speak of cause-effect in social research, but of correlations.

The conclusions should be more developed, establishing lines of reflection on the subject.

Best regards

Author Response

Reviewer 3

I believe that the work presented is interesting although it has been extensively researched and has reached the expected conclusions.

Response: Dear reviewer, we appreciate your comments and try to carry out all the suggestions in the comments. I believe that the innovative role of the study lies in the role of physical activity in the relationship between stress and quality of life for this important professional class, teachers.

The introduction should clarify the different concepts presented, defining them to frame the research.  In this sense, stress and physical activity are mentioned, but they are not defined or framed for a clear understanding.  On the other hand, concepts appear that if they are not developed do not have a clear coherence in the discourse, as is the case of professional burnout.  It would also be interesting, from my point of view, to link the introduction with concepts such as subjective wellbeing.

Response: Dear reviewer, we thank you for your comment and we have introduced the concepts in the introduction, which can also be seen below:

“Stress can be understood as a normal reaction of the body when situations of danger or threat are experienced, causing the body to be in a state of alert [10]”.

“Physical activity is defined as any body movement produced by the musculoskeletal system and where energy expenditure is above resting levels [29]”

In the introduction to make it clearer, we made the link between stress and burnout, as can also be seen below.

“Stress results from a hard situation that threatens homeostasis, which could cause a state of worry or mental tension. Stress can be understood as a normal reaction of the body when situations of danger or threat are experienced, causing the body to be in a state of alert [10]. The risk of stress and burnout related to work seems to increase according to the number of working hours [9]. Teachers with a high level of workload can present sleep problems, burnout, and reported more stress levels in different countries [11–13]. A study carried out with 635 teachers from 47 Americans from September to October 2020 showed that stress levels were high and more prevalent in female teachers [14]. In a nationwide survey in Japan carried out from June to December 2018, it was observed that the high workload in public junior high school teachers was associated with higher levels of stress [15]. Burnout syndrome (or Professional Exhaustion Syndrome) is an emotional disorder with symptoms of extreme exhaustion, stress, and physical exhaustion resulting from exhausting work situations, which demand a lot of responsibility”.

Regarding the link the introduction with concepts such as subjective wellbeing, we tried to make it clearer, especially when addressing quality of life.

Quality of life is a subjective perception and involves the perception of health and well-being considering physical, emotional, and social aspects [21,22].

As for the instruments used, more information should be given on their validity and reliability.  It would have been appropriate to use a mixed methodology to deepen the knowledge of the subject, using focus groups and interviews, as they have proven to be very valid instruments to investigate people's perceptions.

Response: Dear reviewer, thank you for your comment. Regarding the instruments used, we incorporated the suggested changes and inserted them into the methods. As for the focus groups and interviews, unfortunately, the time given to us by the schools to carry out the assessments was little and that is why we were unable to carry out these procedures.

“Quality of life

Quality of life was assessed using the SF-36 questionnaire proposed by Ware et al. [44] and validated for the Brazilian population [45]. This questionnaire has eight quality-of-life domains: functional capacity, physical limitations, body pain, general health status, vitality, social aspects, emotional aspects, and mental health. The score of this instrument ranges from 0-100, where the highest score indicates a better quality of life. In the present study, quality of life was divided into quartiles, and participants located in the highest quartile (Q4) were considered to have a high quality of life, while participants in the lower quartiles were considered to have a low quality of life. This instrument was chosen to be used in the present study due to the different QoL domains it assesses and because it is validated for the Brazilian population.”

Habitual Physical activity

“Habitual physical activity was assessed by the questionnaire of Baecke et al., [46] which was validated against gold standard methods for measuring physical activity, such as accelerometry, in the Brazilian population [47]. This instrument assesses the different domains of physical activity, considering occupational activities (load you have to lift at work; time you need to walk at work; how much you sweat and feel fatigued), leisure-time sports activities (amount of time, frequency weekly and intensity of sports practice) and commuting activities (time spent actively commuting to work, shopping, etc...). The sum of the scores of all domains provides a dimensionless score, where the highest value indicates that the participant is more physically active. Because it is an instrument that broadly assesses physical activity in different domains and can also be translated for the Brazilian population, this questionnaire was chosen for the present study.”.

With respect to the results, I consider that they are unbalanced in relation to the rest of the sections.  They should be developed further.

Response: Dear reviewer, we appreciate your comments. Based on your suggestions, we added more information to the results, such as a figure showing the prevalence of stress among teachers and making the comparisons in Table 1 clearer. The information can be viewed in the results section and below:

“The self-reported prevalence of stress by public school teachers can be considered high, as approximately 50% of teachers reported having moderate stress, while 16.8% reported having intense stress. Only 2.8% report experiencing very intense stress. This information is presented in Figure 1”.

“In Table 1, the variables characterizing the agreement sample are compared with self-reported stress levels (low stress versus high stress). The highest quality of life scores were obtained in the physical limitation and emotional limitation domains in teachers with low stress. There was a significant difference in the domains of physical limitation, general health status, vitality, social aspects, mental health, pain, functional capacity, emotional limitations, all these values being higher in teachers with a low level of stress.”

Table 1. Sample characteristics.

Low stress

High stress

Median (II)

Median (II)

P-value

Age (years)#

46.08 ± 10.24

44.84 ± 10.55

0.396

Physical limitation (0-100)

100.00 (25)

75.00 (75.00)

<0.001*

General health status (0-100)

77 (28.0)

67.0 (30.0)

0.004*

Vitality (0-100)

70 (20)

45.0 (25.0)

<0.001*

Social aspects (0-100)

88.0 (20.0)

63.0 (38.0)

<0.001*

Mental health (0-100)

80.0 (16.0)

60.0 (28.0)

<0.001*

Pain (0-100)

44.0 (9.0)

42.0 (9.0)

0.026

Functional capacity (0-100)

90.0 (30.0)

80.0 (40.0)

0.013

Emotional limitations (0-100)

100.0 (10.0)

100.0 (67.0)

<0.001

Physical activity (Baecke’s score)

7.04 (2.79)

7.48 (2.04)

0.201

#Presented as mean and standard deviation, as data normality was detected.

* Indicates the statistical difference between teachers classified as having low and high-stress levels.

The limitations should appear in a clearer and more developed way.  It should be borne in mind that in no case can we speak of cause-effect in social research, but of correlations.

Response: Dear reviewer, we tried to adjust the limitations according to your suggestions. One of the important aspects to be mentioned was the lack of the focus group or interviews to be carried out with teachers, which could help to better understand the causes of stress levels. Thus, we added this in the limitations.

“Another limitation to be reported is the absence of a focus group in the present study or the conduct of interviews with teachers, which could help to better understand the possible causes of stress and also the self-perception of these professionals regarding this situation.”

The conclusions should be more developed, establishing lines of reflection on the subject.

Response: Dear reviewer, based on your comments and the findings of our study, we have adjusted the conclusion.

“Approximately 70% of teachers participating in this study reported having high levels of stress. Higher levels of stress were associated with seven out of eight domains of the quality of life. However, the physical activity practice attenuated the relationship between higher levels of stress and lower perception of general health status and functional capacity domains of quality of life in secondary public school teachers in a Brazilian city. Therefore, encouraging teachers to practice physical activity can be an alternative for reducing stress levels and improving quality of life. Programs that could contribute to increasing physical activity in the workplace, for example, could be an alternative to be tested for these objectives.”.

Reviewer 4 Report

Comments and Suggestions for Authors

Thank you so much for inviting me to review this manuscript. I leave here some comments that could improve the manuscript. 

- The authors should further improve the need for this study. Why is it important for the scientific literature?

- The authors have to check if the statistical power is adequate for the low number of participants included.

- Did the authors check the assumptions of normality of residuals, linearity, homoscedasticity, multicollinearity, etc.? They did not mention in the statistical analysis section.

- Please replace "Adjusted by" with "Adjusted for".

- However, in model 2, the associations of high 21 stress levels with general health status (P=0.052) and functional capacity (P=0.081) domains of 22 QoL were attenuated. It is not correct. "Mitigated" requires cause-effect relationships. 

- "Our results indicated that PA mitigated the relationship between higher levels of stress and lower perception of general health status and functional capacity domains in public school teachers". Please, be careful with this type of expressions.

- "In addition, p values of 0.050 to 0.100 can be expressed as "some tendency toward significance".

- "Based on the responses, those participants who reported very low or low stress were categorized as low stress, while those who reported moderate, intense, or very intense were classified as having a high level of stress". Please, give a reason.

- "The difference between the individuals classified as having low and high stress levels was analyzed by the Student t-test". Did the authors verify the normality of the variables? Homogeneity of the variants? The statistical analysis section needs to be improved.

Best wishes,

Author Response

Reviewer 4

Thank you so much for inviting me to review this manuscript. I leave here some comments that could improve the manuscript.

Response: We appreciate the reviewer's comment and try to make all suggested changes.

- The authors should further improve the need for this study. Why is it important for the scientific literature?

Response: Dear reviewer, the class of teachers is one of the most important, as these professionals are responsible for the beginning of our literacy and training. However, working conditions and workload can contribute to the development of mental problems, such as stress, and, subsequently, to a decrease in quality of life. Consequently, a series of problems of emotional and cardiovascular origin can be developed. In this sense, the practice of physical activity has been associated with lower stress levels and better quality of life. However, it is not clear in the literature whether the practice of physical activity could attenuate these possible relationships. Based on your comments and those of other reviewers, we have tried to make this clearer in the introduction.

- The authors have to check if the statistical power is adequate for the low number of participants included.

Response: Dear reviewer, thank you for your comment. Our sample calculation considers an alpha error of 5%, and a confidence interval of 95%, requiring 230 participants to have 80% power.

“The sample size calculation considering a confidence interval of 95%, a power of the test of 80%, and a maximum tolerable error of 5%, provided a minimum sample of 230 teachers.” 

- Did the authors check the assumptions of normality of residuals, linearity, homoscedasticity, multicollinearity, etc.? They did not mention in the statistical analysis section.

Response: Dear reviewer, based on your comments, we analyzed the distribution of data (parametric or non-parametric) and presented the information respecting these assumptions. As most of the variables had non-parametric characteristics, to analyze possible associations, we opted for Poisson Regression with robust variance adjustment. This information was adjusted in the statistical analysis.

Data normality was analyzed using the Shapiro-Wilks test. When normality was detected, the Student's t-test was performed for independent samples (Low stress vs High stress) and data were presented as mean and standard deviation. When normality was not detected, the Mann-Whitney test was used and the variables were presented as median and interquartile range. The association between stress level and quality of life domains was analyzed in two multivariate models: model 1 (adjusted for sex, age, and socioeconomic status) and model 2 (model 1 + physical activity level). To perform this association, Poisson Regression with robust variance adjustment was used. The statistical package used was IBM SPSS 25.0 and the statistical significance adopted was 5%.”

- Please replace "Adjusted by" with "Adjusted for".

Response: Done.

- However, in model 2, the associations of high 21 stress levels with general health status (P=0.052) and functional capacity (P=0.081) domains of 22 QoL were attenuated. It is not correct. "Mitigated" requires cause-effect relationships.

Response: We agree with the reviewer and changed the term “mitigated” to “attenuated”.

- "Our results indicated that PA mitigated the relationship between higher levels of stress and lower perception of general health status and functional capacity domains in public school teachers". Please, be careful with this type of expressions.

Response: Dear reviewer, thank you very much for your comment. We have corrected this terminology throughout the text as suggested.

- In addition, p values of 0.050 to 0.100 can be expressed as "some tendency toward significance".

Response: Thanks for the comment.

- "Based on the responses, those participants who reported very low or low stress were categorized as low stress, while those who reported moderate, intense, or very intense were classified as having a high level of stress". Please, give a reason.

Response: Dear reviewer, this categorization was designed for two reasons: the first is because teachers who reported having very little or low stress had the perception of no or almost no stress. Teachers who reported at least moderate stress were included together with teachers who reported intense and very intense stress because they already had the perception of being under stress. Another point is that we use Poisson Regression, which works with dichotomized variables.

- "The difference between the individuals classified as having low and high stress levels was analyzed by the Student t-test". Did the authors verify the normality of the variables? Homogeneity of the variants? The statistical analysis section needs to be improved.

Response: Dear reviewer, we agree with the comment and have improved the statistical analysis section.

“Data normality was analyzed using the Shapiro-Wilks test. When normality was detected, the Student's t test was performed for independent samples (Low stress vs High stress) and data were presented as mean and standard deviation. When normality was not detected, the Mann-Whitney test was used and the variables were presented as median and interquartile range. The association between stress level and quality of life domains was analyzed in two multivariate models: model 1 (adjusted for sex, age, and socioeconomic status) and model 2 (model 1 + physical activity level). To perform this association, Poisson Regression with robust variance adjustment was used. The statistical package used was IBM SPSS 25.0 and the statistical significance adopted was 5%.”